# Impact of a Digital Intervention for Literacy in Depression among Portuguese University Students: A Randomized Controlled Trial

**DOI:** 10.3390/healthcare10010165

**Published:** 2022-01-15

**Authors:** Lersi D. Durán, Ana Margarida Almeida, Ana Cristina Lopes, Margarida Figueiredo-Braga

**Affiliations:** 1Department of Communication and Art, University of Aveiro/DigiMedia, 3810-193 Aveiro, Portugal; marga@ua.pt; 2Entre o Douro e Vouga Hospital Center, 4520-211 Santa Maria da Feira, Portugal; anacristinalopes.sp@gmail.com; 3Department of Clinical Neurosciences and Mental Health, School of Medicine, Porto University, 4200-450 Porto, Portugal; mmfb@med.up.pt

**Keywords:** digital interventions, mental health literacy, audiovisual

## Abstract

Digital interventions are important tools to promote mental health literacy among university students. “Depression in Portuguese University Students” (Depressão em Estudantes Universitários Portugueses, DEEP) is an audiovisual intervention describing how symptoms can be identified and what possible treatments can be applied. The aim of this study was to evaluate the impact of this intervention. A random sample of 98 students, aged 20–38 years old, participated in a 12-week study. Participants were recruited through social media by the academic services and institutional emails of two Portuguese universities. Participants were contacted and distributed into four study groups (G1, G2, G3 and G4): G1 received the DEEP intervention in audiovisual format; G2 was given the DEEP in text format; G3 received four news articles on depression; G4 was the control group. A questionnaire was shared to collect socio-demographic and depression knowledge data as a pre-intervention method; content was then distributed to each group following a set schedule; the depression knowledge questionnaire was then administered to compare pre-intervention, post-intervention and follow-up literacy levels. Using the Scheffé and Least Significant Difference (LSD) multiple comparisons test, it was found that G1, which received the DEEP audiovisual intervention, differed significantly from the other groups, with higher depression knowledge scores in post-intervention stages. The DEEP audiovisual intervention, compared to the other formats used (narrative text format; news format), proved to be an effective tool for increasing depression knowledge in university students.

## 1. Introduction

There are many digital resources that provide mental health information and support. Digital technology has become an addictive element used by young university students as a privileged tool to access information [1]. It is not surprising that young people seek support and information about mental health on the Internet [2,3,4]. However, much of the digital content on the Internet does not have scientific validity [5,6], which can become a problem due to the use of unreliable information.

Providing mental health knowledge and promoting health literacy to young university students is a challenge for universities and the public health system [7,8]. Nowadays, an increasing number of cases of young people with depression are undetected, unrecognized and undertreated, leading to tragic episodes such as suicide and causing a great impact on the family and social environment [9,10,11]. According to the World Health Organization, depression is the leading cause of disability worldwide [12]. University students are exposed to specific challenges given their new responsibilities and are reported as a risk population for mental health problems, namely anxiety and/or depression [13], which can trigger other, more serious, disorders [11]. Young people are often reluctant to seek professional help for a mental/psychological disorder [7,14], due to preconceived ideas imposed by society, low mental health literacy and fear of being exposed [15,16].

Depression, according to Becken [17], is caused by a negative view of the world. A person with depressive symptoms has a negative cognition of the things around them. Depression is an illness in which feelings of deep sadness, emptiness, tiredness and lack of interest are present, which can lead to serious consequences such as suicide, causing great difficulties in family and social contexts [11].

Health literacy is of fundamental importance to guarantee a better quality of life on a personal and social level. This is a process that comprises three fundamental points, namely the capacity to analyze, understand and communicate [16,18], and depends on the skills that the individual or society develops in order to obtain the expected results. The significance of mental health literacy highlights the need to increase knowledge of mental-psychological disorders, so that help and information are sought and stigma is reduced [8,18].

In recent years, digital programs have been developed for the promotion, education and prevention of mental illnesses and/or disorders, with an emphasis on depression [18]. According to Frank, Pong, Asher and Soares [19], the use of digital programs may have positive effects on the understanding of depressive symptoms. The use of digital media to support interventions on depressive symptoms has been the subject of recent studies [20,21], which demonstrates the potential for using technology as a tool for the distribution of digital content in the area of depression [22]. Despite the wide variety of digital resources to treat and prevent depression, only a small number have been validated by specialists in the field [6,23]. For this reason, it is essential to develop further studies to validate the use of digital interventions on depression [24,25], and to better understand how the use of digital resources can provide well-being and mental health for university students.

Digital interventions can be used to provide a set of educational strategies with a cognitive orientation, allowing participants to learn about and/or face situations related to psychiatric disorders [26]. These interventions promote the integration of participants regardless of their geographical location [26,27] and can therefore bring together a large number of people. This makes the learning and/or treatment process more productive, easier and more enjoyable [28]. Digital interventions aimed at mental health promotion and education address specific needs and have a high success rate in overcoming stigma [29]. Importantly, digital resources have great potential for health information provision [30]. Social support, lack of geographic boundaries, free access and ease of access are some of the advantages of digital resources for health promotion and literacy [19,31,32]. However, there are several concerns, such as disparities in Internet access, the quality of online health information and the lack of real support to monitor how this information is processed [1,29,33].

For Michie et al. [34], Hollis et al. [35] and Alkhaldi et al. [36], digital interventions for mental health promotion and care must pay close attention to the content and information to be presented [33,37]. These are special interventions that, because they deal with sensitive issues, must be supported and monitored by specialists to prevent them from being non-beneficial resources for the participants [38]. Similarly, these digital resources for mental health promotion and literacy are based on pedagogical techniques, adapted to the needs of the participants. The information and content developed must have a technological, educational and explanatory context to ensure that the objective of a digital mental health literacy intervention is met [39].

The incorporation of an audiovisual format in the area of mental health literacy is considered an effective strategy to communicate, promote and support mental health literacy [40,41,42].

The evolution of technology and the digital world are part of the general population’s life, especially among young people, thus allowing digital social media to be used as a tool to carry information in the area of mental health [43,44]. The concept of literacy that is associated with the ability to read and write thus expands and becomes the competence to promote or acquire information about, in this case within the scope of mental health [20,45]. The concept of literacy linked to technology is represented through images, sounds and videos, among other things [46]; therefore, the audiovisual format can be defined as a strategy to carry information in different forms of representation that generate interest in young audiences [47].

The digital audiovisual intervention DEEP consists of 23 short videos interspersed between a web series called “The Sara Wound” and informational videos about depression, divided into two stages. The first stage is “DEEP IN”, which exposes the onset and acceptance of depressive symptoms, and the second stage “DEEP OUT”, which presents the phase of seeking help and recovery.

This study aimed to evaluate the impact of the DEEP digital audiovisual intervention on Portuguese university students. The study considered their knowledge about the relevance they should give to symptoms and possible treatments, compared the audiovisual format of the intervention with the narrative text format and the news format and assessed the level of literacy before and after the intervention.

## 2. Materials and Methods

### 2.1. Study Context and Ethical Considerations

This study was conducted as part of the eMental project (evaluation of digital interventions for depression and suicide promotion and literacy), which aims to develop digital interventions for young university students and to understand the role they play in depression and suicide literacy. This research was developed as a randomized controlled trial, and the research protocol was approved by the Ethics Board of the University of Aveiro, Portugal (46-CED/2019).

### 2.2. Study Design and Sample

The evaluation of the impact of the intervention was conducted over a period of 14 weeks with an initial sample of 98 students, aged between 18 and 38 years old, of which 66% were female and 34% were male. The participants were students from two Portuguese universities and were randomly and equally divided into four groups, each group having access to information through different formats during the intervention. It should be noted that Group 1 received the DEEP intervention in audiovisual format and Group 4 was the control group. The purpose of having four groups was to allow comparison of the audiovisual format of DEEP intervention with the narrative text format of the same intervention, the narrative news format and to have a control group. It is important to note that only 71 students completed the first phase of the study.

Full access to the final version of this intervention cannot be presented in this paper as DEEP is still under analysis and development.

### 2.3. Recruitment of Participants

Students from two Portuguese universities were invited to participate in the study by means of an institutional email sent to all students, poster publications on the social networks of the universities’ academic associations and printed posters placed in the common areas of the universities. The only criterion for participation was to be a university student, and the willingness to participate. No exclusion criteria were applied.

### 2.4. Instruments

After the recruitment campaign, those who were interested responded to the email quieroparticipar@ua.pt, sharing their intention to participate in the study. One week afterwards, participants were randomly divided into four groups. All groups were then sent a link via email to the initial questionnaire containing an introduction, an informed consent to participate form, a socio-demographic assessment (age and gender) and the pre-intervention knowledge literacy questionnaire. The literacy questionnaire was adapted from Griffiths et al. [15], Hart et al. [38] and Heickie et al. [48] and was tested in a pilot evaluation [25]. The questionnaire consists of true and false questions, divided into two parts: a first part of 25 questions on symptoms of depression, and a second part with 11 questions related to possible treatments. For the elaboration of the questionnaires and data collection, the software LimeSurvey was used on the platform https://forms.ua.pt/ (accessed in 10 January 2022), from the University of Aveiro, Portugal.

Subsequently, content was sent via email to each group, ranging from DEEP intervention in the digital format for G1, to DEEP intervention in narrative text for G2, to four news items on depression for G3, and it followed a distribution schedule (Appendix A) for a period of 23 working days between 3 p.m. and 8 p.m. At the end of the distribution of content for each group, the literacy questionnaire was sent as a post-intervention measurement instrument.

The purpose of the literacy questionnaire was to characterize participants’ knowledge about depression at pre-intervention, post-intervention and follow-up. Finally, and after receiving the follow-up responses, DEEP intervention was sent in audiovisual format to all groups, including the control group (G4). Figure 1 represents the timeline of the assessment design.

### 2.5. Statistical Analysis

For the quantitative data analysis, IBM^®^ SPSS^®^ software, version 24.0 for Windows^®^, was used to compare the total scores of the four groups in the three evaluation phases (pre-, post-intervention and follow-up), using one-factor ANOVA. When it was verified that there was no normality in the sample, the Kruskal–Wallis test was used as a non-parametric alternative. Since significant differences were found, multiple comparison tests were carried out using the Scheffé and Least Significant Difference (LSD) tests as they are adjusted when there is no normality and homogeneity of variances. For all cases, a level of 5% was used for the statistically significant value (*p* < 0.05).

## 3. Results

The results of the socio-demographic data are shown in Table 1. Considering the 71 university students who completed the entire literacy questionnaire and the socio-demographic data questionnaire in the pre-intervention phase, the age range was from 18 to 38 years old, the largest number of participants were female, and for the marital status of the participants, 60 out of the 71 were single. Regarding the place where they lived during the class period, 51 had to move from the family residence into a university residence due to the geographical distance between their home residences and the university.

It is important to note that the initial sample decreased when comparing each of the phases (pre-, post- and follow-up) of the intervention. Only 36 students reached the end of the study, a drop-out rate of 63.36%. Table 2 shows the number of students per group throughout the study and the results per group of the two literacy questionnaire sections.

To measure the knowledge of respondents in the three phases of the intervention (pre-, post- and follow-up), the literacy questionnaire was used, divided into two sections: “Symptom identification” (Section 1) and “Possible treatments” (Section 2). For each section, the number of correct answers was added together, resulting in a final score. Therefore, the scores for Section 1 ranged from 0 to 25, and for Section 2, the scores ranged from 0 to 11.

It was necessary to test the normality of G1 data in the post-intervention phase, and no statistical significance was obtained. Hence, the assumption that G1 follows a normal distribution was rejected, and therefore, a non-parametric Kruskal–Wallis test was performed to measure if there were significant differences and to determine if the level of knowledge was equal or not in all phases.

In Table 3, the results of the Kruskal–Wallis test can be observed. For Sections 1 and 2 of G1, it can be seen that the significant group differences for each section are under 5%, thus rejecting the hypothesis that knowledge about depression is the same in all three phases.

After rejecting the hypothesis of equality, it was important to know which phase was responsible for this difference by creating a score-ordering variable for multiple comparisons between phases, using the Scheffé test for Section 1 and the LSD (least significant difference) test for Section 2.

The results of these multiple comparison statistical tests are in Table 4. The significant differences are marked with an asterisk, where it can be observed that G1 in the follow-up phase showed higher literacy levels compared to the pre-intervention phase. In Sections 1 and 2 in the pre-intervention phase, G1 had lower literacy levels than in the post-intervention and follow-up phases.

## 4. Discussion

This study evaluated the impact of the DEEP audiovisual intervention on university students. When comparing the results obtained in the pre- and post-intervention phases, it becomes clear that the mean number of correct answers in both sections only increased in groups G1 and G2. Significant differences (*p* > 0.05) were only obtained in G1. Therefore, depression literacy levels were significantly higher in this group, which demonstrates that digital content has a high potential to provide mental health literacy [25,26,33,39,40,49].

It is noteworthy that the knowledge of G3, who received information about depression in narrative notecard format, decreased, and that G4, or the control group, maintained the same knowledge in these phases. These results may suggest that young university students find it easier to obtain information through digital content because of the importance they place on the use of technology. At the same time, based on study findings, it may be inferred that the use of other formats may discourage young university students from acquiring new knowledge [2,8,9,13].

According to Carbonell et al. [1], Horgan and Sweeney [2], Griffiths et al. [15] and Uddin et al. [30], university students are immersed in the world of digital technology, so they are more interested in obtaining information and knowledge when the content is digital, which was corroborated by this study’s findings, since it was shown that the format used for G3 and G4 was not as captivating as the digital format to generate interest in learning. However, for the follow-up phase, 2 months afterward, the scores did not increase. Despite this, the G1 and G2 groups maintained a level of knowledge very close to that of the pre-intervention phase. Although a decrease in knowledge of digital literacy interventions over time has been described in the literature [21,47,50], the DEEP intervention managed to maintain relatively unchanged knowledge levels of depression in the participants in the follow-up phase. It is important to further explore the impact of these results and the factors that influenced them by conducting an evaluation with more students and analyzing the different scenarios for each group.

The G4 or control group increased knowledge between the post-intervention and follow-up phases, but this increase was not statistically significant. This could confirm that when there is the presence of a control group, these groups feel the need to seek information on the topic because they are not involved in the interventions.

This work has several limitations, which must be acknowledged.

One limitation was the recruitment of participants to the study. Despite the strategies used, it was difficult to motivate students to agree to participate. Another possible limitation may have been the use of a questionnaire as a measurement instrument. Although this methodological choice was made in order to measure knowledge during all stages of the intervention, it may have contributed to participants dropping out of the intervention, since many students did not complete the questionnaire in the post-intervention and follow-up phases and therefore gave up continuing in the study. Studies with more than two follow-up phases generally have a high drop-out rate, especially if the participants do not receive a reward that interests them [9,11]. In this study, the drop-out rate was high, representing 63% of the initial sample. In this case, the representativeness of the sample is not fully accomplished, since the final results obtained cannot be generalizable.

However, the drop-out rate during the study did not impede the study, nor did it affect the results, as all four groups maintained an equivalent number of students. Although it is not possible to generalize the conclusions, the results found presuppose an initial step for future studies, in which a strategy should be considered to keep participants enrolled in the study for the duration of the intervention and thus reduce the drop-out rate.

Another limitation of this study was the fact that we could not 100% control the risk of contamination between groups. The choice of participants per group was random; we did not know and could not identify the participants due to the General Data Protection Regulation (GDPR). Alternatives to control the groups were not possible to implement because they would identify the participants; we only checked G1 for the number of views of each video during the time of the intervention.

It could also be considered a limitation of this work that G3 received only four news articles about depression in digital format during the intervention. In fact, this group differed both in content and format from the other groups, but this strategy was used to try to compare the audiovisual format of the digital intervention with the remaining formats, and thus find out if the DEEP intervention in audiovisual format would have more influence on participants’ knowledge of depression.

Youth mental health literacy should be an area of further exploration, so that young people can recognize and respond appropriately to the signs and symptoms of depression or other mental disorders [3,4,9]. Future studies should focus on developing effective technology-linked interventions to improve knowledge and thus raise awareness among young university students about how to care for and maintain good mental health.

## 5. Conclusions

The DEEP digital intervention is based on an audiovisual strategy, grounded in a clinical-social approach, with the aim of improving the depression knowledge of Portuguese university students and with the intention of increasing quality of life and creating a state of full well-being.

The results of this study provide evidence that digital audiovisual content is more likely to increase depression literacy in university students than other formats. Young people learned more from the audiovisual content of the DEEP intervention than students who received the other formats with equivalent information. It is necessary that these interventions are evaluated by specialists before being delivered to the participants, because, as they deal with sensitive topics, information may be shared that is harmful to the participants.

Evaluating the DEEP intervention enriched the perception of the role of digital technologies to promote literacy in depression, highlighting the importance of complementing interventions with two different approaches: information videos (as a substitute for a specialist in the area) interspersed with videos of Sara’s story (portraying the reality of a university student).

Digital technology and mental health together form a key partnership to address current public health challenges and are allies in improving the quality of mental health among university students, which is currently even more fragile due to the pandemic scenario. The DEEP digital intervention format highlighted the potential for videos as a vehicle to increase depression literacy, enabling the understanding of the disease, considering symptom identification and possible treatments. It is also important to note that digital interventions can be scaled up to all audiences and thus provide better health care whether for promotion/literacy/therapy or treatment of mental disorders.

## Figures and Tables

**Figure 1 healthcare-10-00165-f001:**
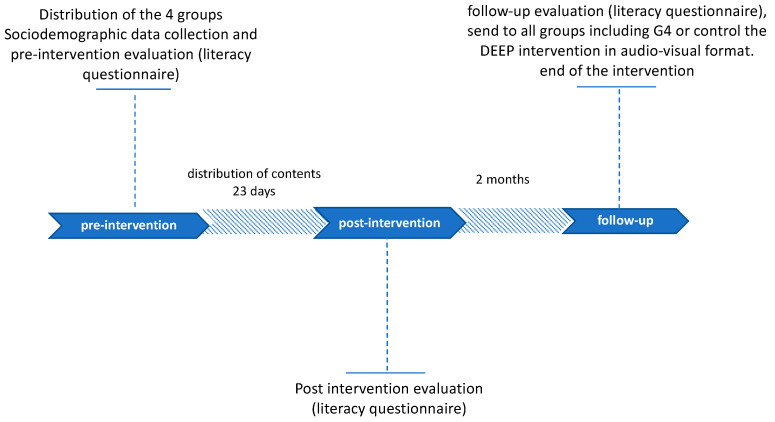
Timeline of the assessment design.

**Table 1 healthcare-10-00165-t001:** Socio-demographic characteristics of participants (*n* = 71).

*n* = 71	Gender		Marital Status			Place of Residence during the Period of University Classes	
Age Range	Male	Female	Single	Married	Partnership	Student Residence	At Home with Their Family
20 and 38 years old	24	47	60	9	2	51	20

**Table 2 healthcare-10-00165-t002:** Results per literacy questionnaire group: number of participants, significant differences, and means in each phase of the study.

Study Group	Pre-Intervention (*n* = 71)				Post-Intervention (*n* = 56)				Follow-Up (*n* = 36)			
	Section 1		Section 2		Section 1		Section 2		Section 1		Section 2	
	Mean	*p* Value	Mean	*p* Value	Mean	*p* Value	Mean	*p* Value	Mean	*p* Value	Mean	*p* Value
G1	20.85	0.056	7.55	0.35	22.19	0.019 *	8.88	0.015 *	20.53	0.095	8.90	0.092
G2	20.82	0.107	6.71	0.451	21.36	0.175	7.14	0.118	20.11	0.264	7.00	0.242
G3	21.81	0.054	7.41	0.262	19.33	0.045	7.33	0.092	18.40	0.043	5.50	0.468
G4	20.41	0.118	7.71	0.094	20.42	0.091	7.50	0.059	20.80	0.191	7.60	0.445

* The mean difference is significant at the 0.05 level.

**Table 3 healthcare-10-00165-t003:** Kruskal–Wallis test results.

Teste	G1	
	Section 1	Section 2
Kruskal–Wallis H	12.367	7.126
*p* value	0.028 *	0.02 *

* The mean difference is significant at the 0.05 level.

**Table 4 healthcare-10-00165-t004:** Results of multiple comparison tests between phases of G1.

Group G1	Group G1	Sheffeé Test			LSD test		
		Section 1			Section 2		
Phase (I)	Phase (J)	Mean Difference (I–J)	Std. Error	Sig.	Mean Difference (I–J)	Std. Error	Sig.
Pre-intervention	Post-intervention	−9.506250	3.872342	0.060	−10.275000 *	4.140478	0.017
	Follow-up	−17.450000	4.471395	0.001	−10.700000 *	4.781013	0.030
Post-intervention	Pre-intervention	9.506250	3.872342	0.060	10.275000 *	4.140478	0.017
	Follow-up	−7.943750	4.653976	0.244	−0.425000	4.976236	0.932
Follow-up	Pre-intervention	17.450000 *	4.471395	0.001	10.700000 *	4.781013	0.030
	Post-intervention	7.943750	4.653976	0.244	0.425000	4.976236	0.932

* The mean difference is significant at the 0.05 level.

## Data Availability

The data that support the findings of this study are available from the corresponding author and the team involved in the project. Data are however available with upon reasonable request and with permission and autorizated the team.

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
