# Peer review of "Impact of a Digital Intervention for Literacy in Depression among Portuguese University Students: A Randomized Controlled Trial"

_healthcare, 2022, doi:10.3390/healthcare10010165_

Round 1

Reviewer 1 Report

This paper provides an evaluation of the impact of a digital intervention for literacy in depression among Portuguese university students. Mental health has become a serious issue especially among college students and audiovisual digital content is an effective way to increase depression literacy in university students.

As the authors have described, young people learn more from audiovisual content. In fact, social networking sites are popular among college students and can include displayed depression references. Also, internet skills, literacy, and how young people utilize technology vary so much by various factors. These are major limitations in this field including this paper, but this does not mean digital intervention is not important to promote mental health. This paper is well written and contains useful information. I have a few minor points the authors may wish to consider.

  1. The authors should write on the limitations in the discussion. The sample size reduces hugely in the follow-up stage. Only 36 students reached the end of the study, with a dropout rate of 63.36%, which raises questions on the reliability of test results.
  2. The generalizability of the present findings may be limited by the self-report questionnaire methodology. However, the study does serve as a starting point in investigating the relationships among the discussed variables.
  3. In the introduction section, one would expect a coherent exposition on mental health knowledge, health literacy, and depression in the definition of key terms.
  4. The literacy questionnaire acronym QL is confusing.
  5. In section 2.4, the acronyms can be written as pre-, post-, and follow-up in place of (T0), (T1), and (T2) which is confusing.

Author Response

Review 1

  1. The authors should write on the limitations in the discussion. The sample size reduces hugely in the follow-up stage. Only 36 students reached the end of the study, with a dropout rate of 63.36%, which raises questions on the reliability of test results.
  2. The generalizability of the present findings may be limited by the self-report questionnaire methodology. However, the study does serve as a starting point in investigating the relationships among the discussed variables.

In relation to comments 1 and 2, the possible causes of the high drop-out rate were written as limitations in the discussion section and because of this, it was not possible to generalize the results found. Therefore, in future work, a strategy to decrease the drop-out rate should be considered and conclusions with more generalized results can be drawn.

  1. In the introduction section, one would expect a coherent exposition on mental health knowledge, health literacy, and depression in the definition of key terms.

In the introductory section, the key concepts of depression, health literacy and mental health literacy were included.

  1. The literacy questionnaire acronym QL is confusing.

The acronym QL literacy questionnaire was removed from the article and substituted by literacy questionnaires.

  1. In section 2.4, the acronyms can be written as pre-, post-, and follow-up in place of (T0), (T1), and (T2) which is confusing.

The acronyms (T0), (T1) and (T2) were replaced by the names of the phases: pre-intervention, post-intervention and follow-up.

Reviewer 2 Report

This is an interesting study, there are some questions:

1- How was the risk of contamination between groups controlled? Are you sure the students in the intervention group did not share the videos with the students in the other groups?

2- What was the reason for placing the third group? This group differs from the intervention group both in content and format and can only be compared with the second group. If the comparison of presenting text and news format was considered by researchers, it seems that this is not in line with the objectives of the current study. It would have been better if the third group had presented different content in digital format.

3- What are the reasons for the high dropout rate in this study? Please explain in the discussion section. It could be useful for other researchers. 

Author Response

Review 2

  1. How was the risk of contamination between groups controlled? Are you sure the students in the intervention group did not share the videos with the students in the other groups?

We could not control 100% the risk of contamination, the choice of participants per group was random, we did not know and could not identify the participants due to the General Data Protection Regulation (GDPR). We tried to find some alternatives to be able to control the groups, but it was impossible, the proposals presented identified the participants in some way, we only tried to control the videos by checking the number of views during the time of the intervention.

  1. What was the reason for placing the third group? This group differs from the intervention group both in content and format and can only be compared with the second group. If the comparison of presenting text and news format was considered by researchers, it seems that this is not in line with the objectives of the current study. It would have been better if the third group had presented different content in digital format.

The reason for placing a third group was that we thought about the possibility of comparing the audio-visual format of the digital intervention with the different formats, namely the narrative text G2, and narrative news G3,  in order to find out if the DEEP intervention in audio-visual format would have more influence on  participants’ knowledge of depression. In terms of content, all formats presented digital content with information about depressive symptoms and possible treatments.

  1. What are the reasons for the high dropout rate in this study? Please explain in the discussion section. It could be useful for other researchers.

 This point was considered in the discussion section and one of the reasons for the high drop-out rate may have been the time between phases (the study took approximately 3 months). Another relevant aspect is that the study had a compulsory questionnaire, as a data collection instrument, and perhaps some participants either did not feel comfortable to respond because it was a sensitive topic, or simply dropped out because they did not want to answer the questionnaires.  

Round 2

Reviewer 2 Report

Thanks for updating the article and answering the questions ,
Therefore, according to your statements, contamination has not been controlled in this study, it is suggested that this case be mentioned as one of the limitations of the research in the discussion section so that the results can be interpreted in the light of the limitations.
The next point is the effect of  the negative news on the third group, which is considered as another interfering factor and should be explained in the discussion.

Author Response

Dear Reviewer
Thank you very much for your comments, and in response we have included in the limitations of the work the suggestions indicated above. 

the authors,
